# Effect of Maternal Nutritional Status and Mode of Delivery on Zinc and Iron Stores at Birth

**DOI:** 10.3390/nu13030860

**Published:** 2021-03-05

**Authors:** Oraporn Dumrongwongsiri, Pattanee Winichagoon, Nalinee Chongviriyaphan, Umaporn Suthutvoravut, Veit Grote, Berthold Koletzko

**Affiliations:** 1Center for International Health, Ludwig-Maximilians-Universitaet Munich, 80336 Munich, Germany; 2Department of Pediatrics, Faculty of Medicine Ramathibodi Hospital, Mahidol University, Bangkok 10400, Thailand; nalinee.cho@mahidol.ac.th (N.C.); u.suthut@gmail.com (U.S.); 3Institute of Nutrition, Mahidol University, Nakhon Pathom 73170, Thailand; pattanee.win@mahidol.ac.th; 4Department of Pediatrics, Dr. von Hauner Children’s Hospital, University Hospital, Ludwig-Maximilians-Universitaet Munich, 80337 Munich, Germany; veit.grote@med.uni-muenchen.de (V.G.); berthold.koletzko@med.uni-muenchen.de (B.K.)

**Keywords:** cord blood zinc, cord blood ferritin, iron deficiency, zinc deficiency, nutrition in pregnancy, pre-pregnancy BMI

## Abstract

Zinc and iron deficiencies among infants aged under 6 months may be related with nutrient store at birth. This study aimed to investigate the association between zinc and iron stores at birth with maternal nutritional status and intakes during pregnancy. 117 pregnant women were enrolled at the end of second trimester and followed until delivery. Clinical data during pregnancy, including pre-pregnancy body mass index (BMI) and at parturition were collected from medical record. Zinc and iron intakes were estimated from a food frequency questionnaire. Serum zinc and ferritin were determined in maternal blood at enrollment and cord blood. Mean cord blood zinc and ferritin were 10.8 ± 2.6 µmol/L and 176 ± 75.6 µg/L, respectively. Cord blood zinc was associated with pre-pregnancy BMI (adj. ß 0.150; *p* = 0.023) and serum zinc (adj. ß 0.115; *p* = 0.023). Cord blood ferritin was associated with pre-pregnancy BMI (adj. ß −5.231; *p* = 0.009). Cord blood zinc and ferritin were significantly higher among those having vaginal delivery compared to cesarean delivery (adj. ß 1.376; *p* = 0.007 and 32.959; *p* = 0.028, respectively). Maternal nutritional status and mode of delivery were significantly associated with zinc and iron stores at birth. Nutrition during preconception and pregnancy should be ensured to build adequate stores of nutrients for infants.

## 1. Introduction

Trace elements are essential for fetal growth and development. During late pregnancy, certain nutrients are transferred across the placenta to provide adequate nutrient stores for utilization during early infancy. Nutrient deficiencies during pregnancy were shown to contribute to pregnancy complications, adverse birth outcomes, and compromised nutrient storage in the newborn [1,2,3]. 

Zinc and iron are essential nutrients for fetal development. Zinc and iron content in fetal body were increasing with increased gestational age. Zinc and iron from maternal circulation are transferred through the placenta to provide these essential nutrients to the fetus, especially during the 3rd trimester. Maternal nutrient status during this period is crucial for the endowment of nutrient in the fetal body [1,3]. Cord blood zinc and iron levels are higher than maternal blood levels during pregnancy, reflecting active transport processes across the placenta [4,5]. Several studies showed a positive correlation of cord blood zinc with birth weight [6,7]. A systematic review and meta-analysis confirmed that both maternal zinc status during pregnancy and cord blood zinc were inversely associated with intrauterine growth restriction and low birth weight [8]. A study in preterm and term infants showed that cord blood zinc was positively correlated with gestational age [6]. As cord blood zinc level is associated with birth weight and gestational age, it may reflect the accumulation of zinc in fetus during prenatal life. 

Cord blood ferritin reflects iron endowment of infants at birth and is associated with infant iron status thereafter [9,10]. Several studies reported a positive association of maternal iron status and cord blood ferritin level. Infants born to anemic or iron deficient mothers had lower cord blood ferritin compared to infants born to non-anemic/iron deficient mothers [11,12,13]. In addition, maternal anemia is also related to infant iron status after birth, but it is not clear whether this is due to low iron stores or poor postnatal diets. Exclusively breastfed infants born to anemic mothers not only had lower cord blood ferritin, but also lower serum ferritin at the age of 14 weeks [12].

During the first 6 months of life, infants receive zinc and iron from breastfeeding and utilize nutrient stores at birth to meet their needs. However, zinc and iron deficiencies were reported among infants aged less than 6 months [14,15], which may be related to insufficient nutrient stores at birth, low intakes of breast milk, or intakes of other foods with low contents of these micronutrients. In this study, we aimed to investigate whether cord blood zinc and ferritin levels were related to dietary intakes and status of zinc and iron status during pregnancy and other maternal and newborn factors at delivery.

## 2. Materials and Methods

This is a prospective observational study, following pregnant women from the third trimester until child delivery. Pregnant women who attended the antenatal care clinic (ANC) at Ramathibodi Hospital, Bangkok, Thailand, were enrolled at 28–34 weeks gestation. Inclusion criteria were singleton pregnancy, plan to deliver the baby at Ramathibodi Hospital, and intention to breastfeed. Exclusion criteria were emergency delivery elsewhere. The study protocol was explained to each participant. Once they understood the protocol and were willing to participate the study, written informed consent was obtained. The protocol was approved by human research ethic committee, Faculty of Medicine Ramathibodi Hospital, Mahidol University (ID 03-60-31) and Ethical Committee, Ludwig Maximillian Universitaet, Munich (Project no. 18-015). All the processes of the study were performed according to the Helsinki Declaration.

This study is a part of the larger study following pregnant women from the third trimester to child birth, and infants until 4 months of age. More details of the study protocol were published elsewhere [16]. The sample size (*n* = 120) was determined in the original study, based on mean zinc intake in breastfed infants (1.00 ± 0.43 mg/day) reported by Krebs et al. [17]. Using the significant level of 0.05 and power of 0.8, the minimum sample size was 64. We estimated the possible drop-off of 45–50%. Therefore, the final sample size for recruitment was 120.

Clinical data during antenatal visits and at delivery were collected from medical records, including pre-pregnancy body mass index (BMI), weight gain during pregnancy, complications of pregnancy (i.e., pre-eclampsia, pregnancy induced hypertension and gestational diabetes (GDM)), supplementation during pregnancy, gestational age at delivery, mode of delivery (i.e., vaginal and cesarean delivery), and infant parameters. Other maternal characteristics, namely, family income and educational attainment were also collected using interview questionnaire. All pregnant women, regardless of age, were screened for GDM. Maternal smoking in Thailand is uncommon, and no participants reported smoking in this study.

Iron supplementation is routinely given to all pregnant women attending ANC, either iron tablet (ferrous fumarate; containing 66 mg of elemental iron), or multivitamin and mineral (Obimin-AZ; containing 20 mg of elemental zinc and 66 mg of elemental iron). The prescription was made by an obstetrician, based on the clinical profiles and drug tolerance by pregnant women. Average daily intakes of zinc and iron supplement were estimated by retrospective recalls of the dose and frequency of supplement ingested during the previous month. Dietary intakes were assessed at enrollment using a semi-quantitative food frequency questionnaires (FFQ) by a skilled nutritionist. The FFQ is a comprehensive food list of commonly consumed foods by pregnant and lactating women in Bangkok, developed from a 3-day dietary record in a previous study done at Ramathibodi Hospital (Dumrongwongsiri et al., unpublished data). Nutrient intakes were calculated using INMUCAL software version 4.0 developed by the Institute of Nutrition, Mahidol University, based on nutrient contents in Thai foods, and expressed as iron or zinc intakes per day.

According to dietary recommended intakes (DRI) for Thais 2020 [18], the recommended zinc intake for pregnant women is 10.8 mg/day. For iron, there is no dietary recommended intake, but universal iron supplementation is recommended for all pregnant women throughout the pregnancy period.

Non-fasting peripheral blood samples were obtained at enrollment. Cord blood samples were obtained by obstetricians at the time of delivery from the placental side without milking. The practice of cord clamping depended on each obstetrician preference, and the data was not obtained in this study. The samples were collected from the cut surfaces of the cords, without placing a catheter into the vessels. Therefore, the samples represent mixed venous and arterial cord blood. Serum from both peripheral and cord blood samples was separated and kept at −80 °C until analysis. Serum zinc was analyzed by flame atomic absorption spectrophotometry (GBC Avanta S, GBC Scientific Equipment Pty Ltd., Dandenong, Australia). Serum ferritin was analyzed by a two-step chemiluminescent microparticle enzyme immunoassay on Architect i2000 SR (Abbott laboratories company, Lake Bluff, IL, USA). All containers and equipment used in blood samples collection were demineralized before used to avoid trace element contamination from environment. Zinc and iron deficiencies among pregnant women were defined as plasma zinc below 10.1 μmol/L [19] and serum ferritin below 15 µg/L, respectively [20]. Hemoglobin (Hb) concentration was taken from the ANC records at the first ANC visit, and at the beginning of the 3rd trimester of pregnancy. Hb below 11 g/dL was defined as anemia [21]. The cord blood ferritin values above 370 µg/L was excluded from the analysis due to the potential of inflammation [10].

Statistical analysis of the data was performed using SPSS version 18 (SPSS Inc. Released 2009 PASW Statistics for Windows, Version 18.0. SPSS Inc., Chicago, IL, USA). Descriptive data were presented as mean ± standard deviation (SD) and *n* (%). Comparison of means was performed by a Student *t*-test and one-way analysis of variance (ANOVA). Comparison of proportions was done by a chi-square test. Difference in Hb concentration and prevalence of anemia during 1st and 3rd trimester were analysed using a paired *t*-test and McNemar test. Correlation was determined by Pearson correlation coefficient. Factors which were correlated with cord blood zinc and ferritin, at *p*-value < 0.1 in univariate analysis, were selected for the multivariate analysis. Multivariate linear regression analysis was performed to determine factors associated with cord blood zinc and ferritin. Possible confounding factors including maternal age, infant sex, gestational age and birth order were included in the regression model. A backward elimination method with removal criteria at a *p*-value <0.10 was used for selecting the final regression models. Regression coefficient (ß) and 95% confident interval are presented. The *p*-value <0.05 was determined as statistically significance.

## 3. Results

One hundred and twenty pregnant women were enrolled at ANC. Three women had emergency deliveries at other hospitals and were excluded from the study. Hence, 117 pregnant women completed the study, and their characteristics and infant parameters are shown in Table 1. The rate of GDM and cesarean delivery in the study population were higher than in other areas of the country. All pregnant women were given iron supplementation, while 109 (93.2%) pregnant women were also given zinc supplements. Dietary zinc intake, without supplementation, was lower than Thai recommended daily allowances. Only 16.8% of pregnant women had adequate dietary zinc intake from the diet alone. When combined with prenatal supplementation, zinc intakes of 92.4% of pregnant women met the recommended daily allowance.

Maternal Hb, serum zinc and ferritin levels, and cord blood zinc and iron levels are shown in Table 2. During the 3rd trimester, the Hb concentration was significantly decreased when compared to Hb concentration at 1st trimester (11.7 ± 1.0 vs. 12.2 ± 1.0, respectively; *p* < 0.001). The prevalence of anemia increased in the 3rd trimester when compared to the 1st trimester (23.9 vs. 10.3 %, respectively; *p* = 0.002). More than half of participants had zinc deficiency (51.3%) while prevalence of iron deficiency (14.5%) was much lower.

Factors associated with cord blood zinc and ferritin levels (unadjusted and adjusted for potential confounders) are presented in Table 3 and Table 4, respectively. Pre-pregnancy BMI was positively associated with cord blood zinc (adj ß 0.15, 95% confidence interval (CI) [0.02,0.28], *p* = 0.023) and negatively associated with cord blood ferritin (adj ß -5.23, 95% CI [−9.14, −1.33], *p* = 0.009). Pregnant women with vaginal delivery had significantly higher zinc and ferritin in cord blood, compared to those who had cesarean delivery (*p*-values 0.007 and 0.028, respectively). Maternal zinc status was significantly related to cord blood zinc (Table 3), but there was no association between maternal iron status (serum ferritin) or Hb and cord blood ferritin (Table 4).

## 4. Discussion

In the population of pregnant women from Bangkok, Thailand that were studied, we found a high prevalence of zinc and iron deficiencies during the 3rd trimester. Intakes of zinc and iron from the diet alone during pregnancy in our study were inadequate to meet the needs, whereas an approximately adequate recommended intake of both nutrients was achieved in women who took prenatal supplementation. Although all women took iron supplements and more than 90% zinc supplements, a high proportion had inadequate levels of blood markers. This might be caused by the change in nutrient requirement and physiology during pregnancy. Recommended intake was established by the evidence of nutrient from dietary intake. The metabolism of zinc and iron in supplementation may differ from the natural nutrient. The appropriate doses of iron and zinc supplementation, to raise the levels of blood markers during pregnancy, needed further investigation. In addition, serum zinc may not be the best biomarker for determination of zinc status in the population.

The benefits of iron supplementation during pregnancy have been shown in both iron-depleted and iron-replete populations as pregnancy progresses [22]. On the contrary, the benefit of zinc supplementation during pregnancy has not been clearly demonstrated. It is estimated that very high proportion (82%) of pregnant women worldwide have inadequate zinc intake [23]. Zinc supplementation during pregnancy is not mandated by the Ministry of Public health in Thailand, but some prenatal supplements for pregnant women marketed in Thailand contain zinc (15–20 mg/d). Zinc intake and status among pregnant women in Thailand should be investigated further to determine whether a general supplementation policy would be advantageous.

Mean (±SD) cord blood zinc and ferritin among the study population were 10.8 ± 2.6 μmol/L and 176.7 ± 75.6 μg/L, respectively. Pre-pregnancy BMI, mode of delivery and maternal status in late pregnancy (28–34 weeks) were associated with cord blood zinc level. Pre-pregnancy BMI and mode of delivery were associated with cord blood ferritin level. We did not find any association of dietary intakes or supplementation with cord blood zinc and ferritin levels.

Our results show that pre-pregnancy BMI and maternal obesity (pre-pregnancy BMI > 23 Kg/m^2^) were positively associated with cord blood zinc level. Previous studies on the relationship between maternal obesity and cord blood zinc showed mixed results. Al-Saleh et al. [24] reported higher cord blood zinc among pregnant women having higher pre-pregnancy BMI; whereas a case-control study of obese and lean pregnant women found no difference of cord blood zinc level between 2 groups [25]. We also found that maternal serum levels of zinc during late pregnancy were associated with cord blood zinc regardless of maternal pre-pregnancy BMI (Table 4). This finding is consistent with the result of a meta-analysis of 23 studies [8]. In addition, this meta-analysis showed that low cord blood zinc level was associated with increased risk of low birth weight. However, we did not find this association, possibly due to a small number of infants with low birth weight (data not shown). The reported associations underline the importance of an adequate zinc status during pregnancy which contributes to the fetal growth and fetal zinc storage. Mechanisms for the effects of zinc on fetal growth need to be further elucidated.

In contrast to zinc, our study showed that pre-pregnancy BMI, but not serum ferritin, was associated with cord blood ferritin level. Pre-pregnancy BMI and maternal obesity were negatively associated with cord blood ferritin and there was no association between maternal iron status or deficiency (based on serum ferritin concentration during the 3rd trimester) and cord blood ferritin in our sample. Several other studies showed significantly lower cord blood ferritin in obese compared to normal weight pregnant women [26,27,28,29]. Hepcidin, which is a regulator of iron homeostasis, was increased in obese pregnant women and associated with poor maternal iron status and impaired iron transfer to infants [30,31,32]. Previous studies found cord blood ferritin associated with iron status and inversely with anemia during pregnancy [9,12,33]. A very large study showed a small but significant correlation (*r* = 0.07 with *p* < 0.001, *n* = 3247) between maternal serum and cord blood ferritin [34], and stronger correlation was demonstrated among pregnant women with serum ferritin below the threshold level of the study (13.6 μg/L). Studies in animal models showed placental adaptation for iron transport in response to maternal iron deficiency or low iron intake [35,36].

Our study also found the mode of delivery associated with both cord blood zinc and ferritin levels. The results show that infants born by vaginal delivery had higher cord blood zinc and ferritin compared to those born bed by cesarean delivery (Table 3 and Table 4). Similar to our results, previous reports found higher cord blood zinc among infants born by vaginal delivery when compared with elective cesarean delivery [37,38]. Cesarean delivery was associated with lower cord blood ferritin [39]. Other micronutrients such as copper, magnesium and manganese were also found to be higher in cord blood among pregnant women with vaginal delivery compared to elective cesarean delivery [37,38]. This might be explained by a lower placental transfusion during active labor among pregnant woman with cesarean section compared to vaginal delivery. This was demonstrated in a study showing a higher placental residual volume and lower Hb concentration in cord blood of a mother who has had a caesarian delivery [40].

There was no association between dietary intake and cord blood nutrient levels in our study. To the best of our knowledge, there is no published study showing a correlation between zinc intake during pregnancy and cord blood zinc. For iron, a randomized control trial of iron supplementation in pregnant women showed no difference in cord blood ferritin concentration between placebo and supplemental group [41]. An isotope study of iron transfer to the fetus during the 3rd trimester of pregnancy found a different iron transfer to the fetus between pregnant women with and without iron supplementation, but there was no difference between cord iron profiles [42]. Prenatal iron supplementation might be more critical to maintain maternal iron status and related functions, while there is less impact on iron transfer to the fetus. Therefore, to ensure an adequate iron status in early infancy, other measures may be needed such as delayed cord clamping, which was shown to effectively increase infant iron stores [43].

To the best of our knowledge, the present study is the first in Thailand investigated zinc and iron stores of infants at birth. The study results emphasized the significance of maternal nutrition which influences nutrition during early life. As we found pre-pregnancy BMI was related to zinc store of infants at birth, this demonstrated the importance of pre-conception nutrition status of women at reproductive age. The natural mode of delivery had some advantages on nutrient transfer during delivery.

There were some limitations in our study. Cord clamping methods, which are related to iron transfer during delivery, were not recorded in our study. Data regarding prenatal supplementation may be prone to some error because they were retrospectively recalled by participants, and the actual compliance was not monitored. Further studies should investigate the effect of maternal nutritional status and prenatal supplementation on micronutrient status of infants after birth.

## 5. Conclusions

Zinc and iron deficiencies were highly prevalent among pregnant women in this study. Dietary intakes during pregnancy were problematic. Zinc storage of infants was related with maternal zinc level during pregnancy. Pre-pregnancy BMI was associated with cord blood zinc and ferritin concentrations. Vaginal delivery may provide an advantage for micronutrient transfer during active labor. Maternal nutrition during preconception and pregnancy affects some of the nutrient stores in infants. Prenatal iron supplementation has been universally recommended where iron deficiency is highly prevalent to prevent serious consequences of maternal death associated with child birth. In addition to iron supplementation, zinc supplementation may be needed in light of the high prevalence of zinc deficiency during pregnancy, and its significant contribution to the zinc store as reflected by cord blood zinc. Further study is needed to assess the magnitude of zinc deficiency in the population and effective intervention to improve both maternal zinc status and the infant store.

## Figures and Tables

**Table 1 nutrients-13-00860-t001:** Characteristics of study participants (*n* = 117)

Characteristics	Mean ± SD	*n* (%)
Before & During pregnancy		
Maternal age (years)	31.9 ± 5.6	
Pre-pregnant BMI (Kg/m^2^)	22.0 ± 3.9	
<18.5		21 (17.9)
18.5–22.9		52 (44.5)
≥23		44 (37.6)
Zinc intake (mg): Diet Supplement Diet and supplement Iron intake (mg): Diet Supplement Diet and supplement	8.5 ± 3.0 17.2 ± 5.8 25.7 ± 6.8 11.2 ± 5.0 63.8 ± 19.1 74.9 ± 20.1	
Education level		
high school		20 (17.1)
bachelor degree		74 (63.2)
higher		23 (19.7)
Family income (baht/month)		
<10,000		5 (4.3)
10,000 – <30,000		39 (33.3)
>30,000		73 (62.4)
Birth order		
1st child		70 (59.8)
2nd child		41 (35.1)
3rd child		6 (5.1)
Complication of pregnancy		
gestational diabetes (GDM)		25 (21.4)
pregnancy induced hypertension/preeclampsia		1 (0.9)
Mode of delivery		
vaginal delivery		63 (53.8)
cesarean delivery		54 (46.2)
Birth parameters		
Gestational age at delivery (weeks)	38.5 ± 1.2	
Birth weight (g)	3095 ± 394	
Birth length (cm)	49.7 ± 2.1	
Low birth weight (<2500 g)		8 (6.8%)
Infant sex		
Male		63 (53.8)
Female		54 (46.2)

**Table 2 nutrients-13-00860-t002:** Maternal zinc and iron/anemia status during pregnancy and cord blood zinc and ferritin levels

Laboratory Parameters	Total N	Mean ± SD	*n* (%)
During 1st trimester of pregnancy:			
Hb, g/dL	116	12.2 ± 1.0	-
Prevalence of anemia, % ^1^	116	-	12 (10.3%)
During 3rd trimester of pregnancy:			
Hb, g/dL	117	11.7 ± 1.0	-
Prevalence of anemia, % ^1^	117	-	28 (23.9%)
Prevalence of iron deficiency, % ^3^	117	-	17 (14.5%)
Prevalence of iron deficiency anemia, % ^4^	117	-	6 (5.1%)
Serum zinc, μmol/L	117	11.1 ± 4.8	-
Prevalence of zinc deficiency, % ^2^	117	-	60 (51.3%)
Serum ferritin, μg/L	117	32.3 ± 21.1	-
Cord blood			
Cord blood zinc, μmol/L	114	10.8 ± 2.6	-
Cord blood ferritin, μg/L ^5^	105	176.7 ± 75.6	-

Hb: Hemoglobin; ^1^ Hb < 11.0 g/L, ^2^ serum zinc < 10.1 μmol/L, ^3^ serum ferritin < 15 μg/L, ^4^ Hb < 11.0 g/L and serum ferritin < 15 μg/L, ^5^ Total cord blood samples for ferritin = 110, excluded cord blood ferritin > 370 μg/L (n = 5).

**Table 3 nutrients-13-00860-t003:** Factors associated with cord blood zinc

Factors	Unadjusted Model		Adjusted Model	
	ß (95%CI)	*p*-Value	ß (95%CI)	*p*-Value
Pre-pregnancy BMI	0.06 (−0.05, 0.19)	0.30	0.15 (0.02, 0.28)	0.023 *
Mode of delivery ^1^	1.12 (0.18, 2.06)	0.020 *	1.38 (0.38, 2.38)	0.007 *
Maternal zinc status	0.10 (0.005, 0.20)	0.039 *	0.12 (0.02, 0.21)	0.023 *
Maternal age	−0.05 (−0.13, 0.04)	0.28	−0.05 (−0.14, 0.04)	0.30
Birth order ^2^	−0.08 (−0.90, 0.72)	0.84	−0.05 (−0.91, 0.81)	0.90
Gestational age at birth	−0.05 (−0.44, 0.35)	0.80	−0.16 (−0.56, 0.23)	0.41
Infant sex ^3^	−0.04 (−1.01, 0.92)	0.92	−0.32 (−1.28, 0.64)	0.51
Dietary zinc intake	−0.05 (−0.21, 0.11)	0.53	−0.09 (−0.25, 0.07)	0.28

* *p* < 0.05; ^1^ Mode of delivery; 0 = cesarean delivery, 1 = vaginal delivery; ^2^ Birth order; 1 = first child, 2 = second or more child; ^3^ Infant sex; 1 = male, 2 = female.

**Table 4 nutrients-13-00860-t004:** Factors associated with cord blood ferritin

Factors	Unadjusted Model		Adjusted Model	
	ß (95%CI)	*p*-Value	ß (95%CI)	*p*-Value
Pre-pregnancy BMI	−5.62 (−9.32, −1.92)	0.003 *	−5.23 (−9.14, −1.33)	0.009 *
Mode of delivery ^1^	50.31 (22.54, 78.07)	0.001 *	32.96 (3.68, 62.24)	0.028 *
Gestational age at birth	10.84 (−0.90, 22.58)	0.07	9.44 (−2.10, 20.98)	0.11
Maternal age	−1.75 (−4.44, 0.95)	0.20	0.53 (−2.24, 3.29)	0.71
Birth order ^2^	−24.76 (−49.40, −0.12)	0.049 *	−9.23 (−35.07, 16.61)	0.48
Infant sex ^3^	28.91 (−0.10, 57.91)	0.05	−0.32 (−1.28, 0.64)	0.51
Maternal serum ferritin	0.20 (−0.48, 0.87)	0.56	0.06 (−0.59, 0.71)	0.86
Maternal Hb during 1st trimester	−0.88 (−15.72, 13.96)	0.91	−3.27 (−17.63, 11.09)	0.65
Dietary iron intake	−0.38 (−3.33, 2.58)	0.80	−0.41 (−3.21, 2.40)	0.77

* *p* < 0.05; ^1^ Mode of delivery; 0 = cesarean delivery, 1 = vaginal delivery’; ^2^ Birth order; 1 = first child, 2 = second or more child; ^3^ Infant sex; 1 = male, 2 = female.

## Data Availability

Presented data are available on request from the corresponding author.

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
