# Peer review of "Effect of Maternal Nutritional Status and Mode of Delivery on Zinc and Iron Stores at Birth"

_nutrients, 2021, doi:10.3390/nu13030860_

Round 1

Reviewer 1 Report

Abstract Lines 15, 20, 22: If women were enrolled during second trimester how did the researchers document pre-pregnancy BMI? This is discussed later in the study design, however it needs to be briefly mentioned in the abstract

Line 19: correct the spelling of "enrollment"

Line 30: Introduction section may benefit from discussion of the developmental stage at which zinc and iron are most critical for fetal development. Is the absorption of iron and zinc constant throughout fetal development? Does the fetus need more iron and zinc during the third trimester? These question need to be addressed to justify the rationale behind the study

Lines 189-192: This sentence is confusing, please explain clearly why iron and zinc levels in the blood were inadequate despite all women were supplemented with iron and zinc during pregnancy

Line 209, 224: pre-pregnancy BMI and maternal obesity were (change from 'was' to 'were')

Lines 208-236: The main finding in the study is that pre-pregnancy BMI and maternal obesity were positively associated with cord blood zinc level and negatively associated with cord blood ferritin. This suggests that mode of transport of zinc and iron are affected in an opposite way by maternal body composition. Possible underlying reasons of this effect need to be discussed. It also needs to be addressed that this is "correlation" and it does not always translate into "causation", it could be merely a coincidence.

Author Response

Thank you for your comment. 

I  have correct some spelling and gramma mistakes in the manuscript as your suggestions.

I added some information in the abstract regarding the pre-pregnancy BMI and edited the abstract to maintain 200 words length.

Information regarding zinc and iron transfer through placenta for fetal development was added in introduction part. 

Additional explanation while participants in this study still had zinc and iron deficiency despite supplementation was added. 

These 2 paragraphs explained the possible mechanism behind the association of pre-pregnancy BMI with cord blood zinc and ferritin. The correlation of pre-pregnancy BMI and cord blood zinc was not consistency and showed mixed results from previous studies. In contrast, the negative association of pre-pregnancy BMI and cord blood ferritin was explained by increased of hepcidin.

Reviewer 2 Report

Comments to the Authors of manuscript number: nutrients-1127498  entitled “Effect of maternal nutritional status and mode of delivery on zinc and iron stores at birth”.

It is known that there is a link between maternal diets and birth outcomes, which shows long-lasting effects in offspring life. Authors used a food-frequency questionnaire is a self-report method commonly used in large-scale epidemiological studies. The study is interesting and is suitable for Nutrients.

  1. L 15 the period of pregnancy in mother is the prenatal period for fetus. It is the same time.
  2. L 25 – the nutrition during preconception time was not included in the study
  3. L 36 this sentence is consistent with L 31
  4. L 37 – this sentence should be rephrased because now it indicates that the concentration of Zn and Fe in cord blood is higher compared to blood level of mother, as a blood volume.
  5. L 44 - the terminology should be uniformed – intrauterine or prenatal
  6. L 52- at the age of..
  7. L 58 – the term of these nutrients suggest that that mentioned in this sentence, while here are listed zinc and ferritin. Thus, there should be listed zinc and iron instead the term "these nutrients"
  8. L 16 “enrolled at second trimester”it is not the same as L 61 “from the third trimester”
  9. L 85-87 – the L 87 is sufficient.

10.L 95 - by whom the questionnaire was developed and validated?

  1. Table 2 – SI units should be used

Author Response

Thank you for your comments.

I have revised the manuscript according to your suggestions. 

I added the information regarding the food frequency questionnaires used in this study. The questionnaires was developed by the same group of researchers at Ramathibodi Hospital in previous research project aimed to study zinc and iron in lactating mothers. 

Reviewer 3 Report

The submitted article "Effect of maternal nutritional status and mode of delivery on zinc and iron stores at birth" has an interesting theme and is well elaborated. I believe that it should be published in this form.

Author Response

Thank you for your review.